# Fine-tuned In-Context Learners for Efficient Adaptation

## Abstract

When adapting large language models (LLMs) to a specific downstream task, two primary approaches are commonly employed: (1) prompt engineering, often with in-context few-shot learning, leveraging the model's inherent generalization abilities, and (2) fine-tuning on task-specific data, directly optimizing the model's parameters. While prompt-based methods excel in few-shot scenarios, their effectiveness often plateaus as more data becomes available. Conversely, fine-tuning scales well with data but may underperform when training examples are scarce. We investigate a unified approach that bridges these two paradigms by incorporating in-context learning directly into the fine-tuning process. Specifically, we fine-tune the model on task-specific data augmented with in-context examples, mimicking the structure of k-shot prompts. This approach, while requiring per-task fine-tuning, combines the sample efficiency of in-context learning with the performance gains of fine-tuning, leading to a method that consistently matches and often significantly exceeds both these baselines. To perform hyperparameter selection in the low-data regime, we propose to use prequential evaluation, which eliminates the need for expensive cross-validation and leverages all available data for training while simultaneously providing a robust validation signal. We conduct an extensive empirical study to determine which adaptation paradigm - fine-tuning, in-context learning, or our proposed unified approach offers the best predictive performance on a concrete data downstream-tasks.

## 1 Introduction

The rapid progress in Large Language Models (LLMs) has unleashed a flood of new techniques for adapting these powerful models to specific downstream tasks. A wide spectrum of approaches have been proposed, from established methods like full fine-tuning or parameter-efficient fine-tuning to a diverse array of prompting techniques, including in-context learning, retrieval-augmented generation, and beyond (Schulhoff et al., 2024). Irrespective of the chosen adaptation strategy, the availability of ground-truth or "golden" data remains essential, not only for evaluating the efficacy of the chosen method, but also for tuning relevant hyper-parameters. Within this landscape, fine-tuning and in-context few-shot learning (Brown et al., 2020) can be seen as two cornerstone techniques for adapting LLMs. In the former, the model's parameters are directly optimized on task-specific data, enabling it to learn nuanced patterns and potentially achieve superior performance with sufficient training data. In contrast, in-context learning leverages the model's pre-trained knowledge and generalization capabilities by providing a few illustrative examples within the prompt, offering a sample-efficient approach particularly suitable when data is limited. However, in-context learning's effectiveness is intrinsically tied to the generalization abilities of the model and is generally expected to not scale as effectively with larger datasets and number of examples in-context (Wang et al., 2023; Li et al., 2024).

We here investigate a unified approach that bridges these two paradigms. Instead of fine-tuning on individual examples, we fine-tune a model on $k$-shot in-context prompts. During inference, the fine-tuned model is again supplied with $k$ in-context examples from the training set in addition to the test-point. We conduct an extensive empirical study comparing the sample efficiency of these approaches across a range of downstream tasks and model sizes. Prior work has explored training on k-shot ICL examples, however generally from a meta-learning perspective (Min et al., 2022; Chen et al., 2022).

We focus on a different scenario: **Given a concrete downstream task, which paradigm offers the best predictive performance, particularly when task-specific data is limited?**

All approaches rely on adequately chosen hyperparameters. For ICL, these relate to prompt construction and response parsing, while for FT, they involve factors like learning rate and number of training epochs. While the literature and established best practices offer reasonable starting points, optimizing these hyperparameters for a specific task often yields significant performance gains. However, the challenges of hyperparameter selection in few-shot and small-data regimes are often overlooked. Prior work frequently tunes hyperparameters on large held-out sets, a luxury not available in data-scarce scenarios. To address this, we propose a hyperparameter tuning protocol that is both data- and computationally efficient, allowing for a fair comparison of ICL and fine-tuning approaches under realistic data constraints. Our protocol, grounded in prequential evaluation, avoids sacrificing valuable ground-truth data for hyperparameter selection alone, while providing a principled and robust assessment of model performance.

We summarize the paper's contributions as follows.

1. We propose to adapt LLMs to specific downstream tasks by fine-tuning on $k$-shot in-context learning prompts, combining sample efficiency of in-context learning with the flexibility and scaling of fine-tuning.

2. We introduce a practical and efficient hyperparameter tuning protocol based on prequential evaluation, eliminating the need for separate validation sets and enabling effective fine-tuning in data-scarce scenarios.

3. We conduct an extensive empirical study to investigate the sample efficiency of in-context learning, standard fine-tuning, and our proposed approach across various model sizes and data-set scales.

## 2 RELATED WORK

Training on $k$-shot in-context learning examples has been explored previously, for example by Min et al. (2022) and Chen et al. (2022). However, these works take a meta-learning perspective and first train a model on a diverse set of ICL tasks, and then apply the frozen model to an unseen task at test-time. The authors demonstrate that such ICL-trained models exhibit improved performance on unseen tasks when provided with k-shot examples. Similarly, Wei et al. (2023) train on a meta-distribution of ICL tasks, however propose to remap meaningful label identifier to random, uninformative identifiers to force the model to rely on in-context information instead of relying on in-weights knowledge aquired during pre-training. We here instead focus on the converse scenario: Faced with a specific downstream task with only relatively few ground-truth examples, how can we best adopt the model for our purpose? Concurrent work by Zhu et al. also explores leveraging in-context information during gradient-based training through 'context-enhanced learning'; however, their method specifically avoids computing autoregressive gradients on the contextual helper material, unlike our ICL+FT approach, which fine-tunes directly on k-shot prompts where the entire prompt structure informs the gradient updates.

Numerous works have investigated the sample efficiency of ICL, with some recent studies extending to the many-shot regime with hundreds or even thousands of examples (Bertsch et al., 2024; Agarwal et al., 2024). However, few studies directly compare ICL to fine-tuning approaches Liu et al. (2022) and Mozes et al. (2023) highlighted the effectiveness of fine-tuning in the few-shot regime when introducing parameter-efficient fine-tuning (PEFT) methods, while Mosbach et al. (2023) recently presented a comparison of ICL and fine-tuning. Prompt- and prefix- tuning (Li and Liang, 2021) approaches, as well as modern variants thereof (Lu et al., 2025) also show the continued effectiveness of gradient based methods especially when the number of available ground-truth exampels exceeds a hand full(Wang et al., 2023; Li et al., 2024).

Another line of research reveals scenarios where context-based approaches can outperform fine-tuning in terms of generalization. Berglund et al. (2023) showed that pretrained LLMs struggle to generalize simple relations learned via gradient updates (e.g., "A=B" to "B=A"), but generalize effectively when such information is presented in context. Similarly, Allen-Zhu and Li (2023) and

---

**Algorithm 1:** *ICL+FT* Prequential Training and Evaluation.

**Input** : Dataset $\mathcal{D} = \{(x_i, y_i)\}_{i=1}^{N}$ (N training examples)
        Number of in-context examples $K$
        Gradient steps per example (number of epochs) $E$
        Initial model parameters $\theta_0$
**Output** : Model parameters $\theta_N$
        Avg. prequential performance metric $L_N/N$

1 Initialize cumulative loss $L_0 \leftarrow 0$ ;
2 **for** $i = 1$ **to** $N$ **do**
3     Sample $\min(K, i)$ context examples ctx $\sim \mathcal{D}_{<i}$
4     $\hat{y}_i \sim p(\cdot \mid \text{ctx} \oplus x_i, \theta_{i-1})$ ;             // Predict $y_i$ given $x_i$ and context
5     $l_i = \text{loss}(\hat{y}_i, y_i)$ ;                  // Calculate next-step loss
6     $L_i = L_{i-1} + l_i$ ;                  // Update cumulative loss
7     **for** $1$ **to** $E$ **do**
8         Sample $\min(K, i)$ context examples ctx $\sim \mathcal{D}_{<i}$
9         $\theta_i = \theta_{i-1} - \nabla_\theta \log p(\text{ctx} \oplus x_i \oplus y_i \mid \theta_{i-1})$ ;     // Gradient step
10     **end**
11 **end**

---

Wang et al. (2024) studied related limitations of storing and accessing factual knowledge in the weights of transformers trained from scratch. These findings suggest complementary strengths and weaknesses between gradient-based adaptation and context-based approaches. We here study fine-tuning on k-shot ICL examples and consider this an initial step towards unifying these learning paradigms, aiming to leverage the advantages of both.

## 3 PREQUENTIAL HYPERPARAMETER SELECTION

In this paper, we consider the data-limited scenario, where hyperparameter selection presents a significant challenge. In this regime, standard train/validation splits can become problematic, leading to unreliable results. (K-fold) cross-validation, while suitable for stateless methods like ICL, requires many repeated training runs for FT approaches and is thus computationally expensive and unwieldy to use. We propose instead to draw inspiration from prequential analysis for data-efficient hyperparameter selection.

Prequential evaluation, also known as forward validation, boasts a rich history in statistical learning (Dawid, 1984; Philip Dawid and Vovk, 1999; Grünwald, 2007; Grünwald and de Rooij, 2005). While prequential approaches have found some applications in deep learning, for example in neural architecture search and hyperparameter selection (Lyle et al., 2020; Ru et al., 2021), identifying (causal) structure (Bornschein et al., 2023), evaluating visual feature extractors (Li et al., 2023), and as a heuristic for characterizing the efficiency of in-context learning (Elmoznino et al., 2025), they do not enjoy the same popularity as cross-validation, whose comparative sample-inefficiency is not typically a barrier for the scale of problems on which deep neural networks are trained.

Prequential evaluation is a method for statistical model selection. It works by decomposing the log probability of a data sequence, $x^n = x_1, \ldots, x_n$, into a sum of sequential predictive probabilities:

$$\log p(x^n) = \sum_{i=0}^{n} \log p(x_{i+1}|x^i) . \tag{1}$$

Under this framework, a good model is one that consistently assigns a high likelihood to the next observation throughout the learning process. This demonstrates both data efficiency and strong final predictive performance. This approach is justified by principles like the minimum description length (MDL) (Bornschein et al., 2023) and its relation to the Bayesian marginal likelihood (van der Wilk et al., 2018).

We apply this perspective to hyperparameter selection by following an iterative procedure: Each available data point serves first as a test point for performance assessment and is then incorporated into the training data. Once an observation is considered training data, we use it to compute gradient based parameter updates, and as in-context examples for conditioning. This yields two main benefits: first, it is highly data-efficient as every data point contributes to the hyperparameter selection

criterion; second, it is highly flexible in that once a data point has been evaluated we can train on it for as many additional epochs as is required. Our implementation, for example, performs $E$ gradient steps on $k$-shot ICL sequences sampled from all previously seen examples for each newly evaluated example. Algorithm 1 provides a detailed procedural outline of our method.

Prequential evaluation is not only sample-efficient, but also computationally efficient. When using log-likelihood as the evaluation metric, the loss computation step is effectively free, as it is computed during the first gradient step with each new example. In contrast, evaluation with separate validation sets can be considerably more expensive. For instance, in our experiments, the test-set evaluations (typically on O(100) examples repeated at 6 training set size positions) often dominated the overall computational cost, typically accounting for 1/2 to 2/3 of the overall runtime. See Appendix A for more details and background on the prequential approach.

The prequential performance is a well-motivated metric for model and hyperparameter selection, but it can not be directly compared to the expected performance on held-out data, as it also incorporates the model's performance early during training on few examples only. We here use the prequential approach for fine-tuning and for hyper-parameter selection; all reported metrics are then however computed on held-out test sets. The combination of in-context learning and fine-tuning (*ICL+FT*) that we propose can be employed independently of the prequential approach (see Section 4), however, we find that in practice, these methods complement each other favourably. After a single prequential training run we obtain both a performance metric for model selection and the final model parameters, ready for deployment.

## 4 EXPERIMENTS

Our empirical study comprises two parts. First, we conduct an extensive investigation into the sample efficiency of *ICL-Only*, *FT-Only*, and the combined *ICL+FT* approach across a diverse set of downstream tasks. These experiments show that fine-tuning based methods often outperform *ICL-Only* with as few as 10 to 100 examples, and that *ICL+FT* consistently matches or exceeds the performance of the better-performing individual method. Secondly, we perform a series of ablation studies to shed light on the properties, strengths, and weaknesses of ICL, FT and the prequential hyper-parameter selection scheme.

Our experiments utilize the open-source Gemma-2 models (Gemma Team, 2024), available in 2B, 9B, and 27B parameter sizes. We employ Adafactor (Shazeer and Stern, 2018) as the optimizer due to its reduced memory footprint compared to Adam. Unless otherwise specified, we consistently explore the hyperparameter space: learning-rate $\in \{$1e-4, 2e-4, 3e-4$\}$ and number of epochs $E \in \{$1,2,5,10,15$\}$. For both *FT-Only* and *ICL+FT* we adopt the prequential training and hyperparameter selection approach. The hyperparameter configuration yielding the highest prequential performance is selected and subsequently evaluated on the held-out test data, ensuring no information leakage. For robustness, each experiment is repeated 5 times with different random seeds and we report the $2\sigma$ standard error of the mean ($\approx 95\%$ confidence interval) on all metrics. To investigate sample efficiency, we often subsample larger datasets. To simulate realistic scenarios, and to minimize variance in hyperparameter comparisons, this subsampling process is deterministic, controlled by the random seed. This ensures consistent training and test data across different configurations for a given seed. Consequently, the sample-efficiency plots visualize the expected performance when training on datasets of varying sizes, while all model and parameter selection procedures operate under concrete, data-constrained conditions. If not mentioned otherwise we do not provide any instruction prefix or otherwise manually constructed prompt to these models. For *FT-Only* we train straight on $x \oplus y$ pairs, where $x$ denotes a query, $y$ the desired response and $\oplus$ string concatenation. For *ICL+FT* we prefix the newest training point with the $k$-shot context examples $\Pi_{i=1}^{K}$ "Next example: " $\oplus x_i \oplus y_i$. For training we place a loss and compute gradients based on all responses $y_i$ in the sequence. $K$ therefore also acts similar to a batch-size parameter. For evaluation, we omit the ground truth label $y$ and sample a continuation $\hat{y}$ from the model. We then compare $y$ and $\hat{y}$, ignoring leading/trailing whitespace and other idiosyncrasies of instruction-tuned models. For classification tasks, we use the MAP prediction $\hat{y} = \arg\max_y p(y|\cdots)$ instead of sampling. See Appendix B for details, example prompts and training sequences.

| Method | Model Size | | |
|---|---|---|---|
| | 2B | 9B | 27B |
| *FT-Only* | 27.3% | 52.8% | 62.6% |
| *ICL-Only* | 37.2% | 57.1% | 64.6% |
| *ICL+FT* | **55.3%** | **68.7%** | **72.3%** |

Test-set accuracy after using 30 examples. *ICL-Only* Gemini 1.5 Pro achieves **72.8%**

| Method | Model Size | | |
|---|---|---|---|
| | 2B | 9B | 27B |
| *FT-Only* | 50.7% | 69.7% | 72.6% |
| *ICL-Only* | 37.6% | 56.6% | 64.2% |
| *ICL+FT* | **67.5%** | **78.7%** | **81.8%** |

... after using all but 100 examples. *ICL-Only* Gemini 1.5 Pro achieves **76.6%**

Table 1: Average test-set accuracy over 23 BBH tasks. **Left**: after using 30 or, **right**, all but 100 ground truth examples (= 150 examples for most tasks). For ICL-Only we report the best performing $\{1, 3, 5, 10, 15, 30, 100\}$-shot performance. Note that *ICL+FT* with a smaller models often outperforms *ICL-Only* larger models by considerable margin.

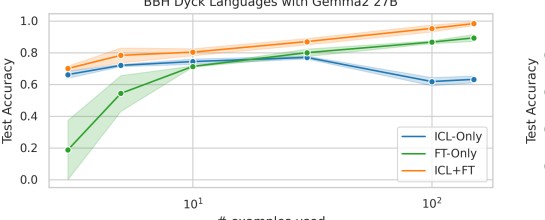
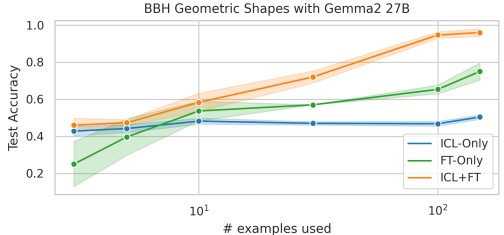

Figure 1: Two representative results from the Big-Bench-Hard suite with between 3 and 150 ground truth examples. We observe the typical signature where *ICL+FT* performs on-par with *ICL-Only* when very few examples are available, but improves similarly or better than *FT-Only* with more training data. Appendix C.1 shows the individual results on all 23 BBH tasks.

**Big Bench Hard.** We conduct an extensive methods comparison on the Big Bench Hard dataset (BBH, Suzgun et al., 2022), comprising of 23 tasks identified as challenging for state-of-the-art LLMs in 2022. Most tasks provide 250 examples, with the exception of *Causal Judgment*, *Penguins in a Table* and *Snarks*, which only provide 187, 146 and 178 examples respectively. We emphasize that the prequential training and evaluation protocol described in Section **??** does not necessitate a separate held-out set, allowing practitioners to utilize all data points for training. However, for the purpose of this report we reserve 100 examples from each BBH task exclusively for reporting test-set accuracy. As a result, for most tasks we have 150 examples available for training and hyperparameter selection. We evaluate *ICL-Only* , *FT-Only* and *ICL+FT* with all three Gemma model sizes.

To investigate sample-efficiency we apply these methods to subsets of 3, 5, 10, 30, 100 and the full set of (usually) 150 examples. Figure 2 shows typical results for BBH Dyck Languages and Geometric-Shapes. Appendix C.1 presents the full suite of results. *ICL+FT* is for almost all model sizes and sample-size budgets the best performing approach, or it shares that position with the better one of *ICL-Only* or *FT-Only* within their mutual uncertainty bands. Typically, *ICL+FT* performs on-par with *ICL-Only* when very few examples are available, but improves with the availability of more data analogously to

Table 2: Avg. held-out accuracy on 11 NLP tasks. Wei et al. (2023) report 84.4% with their proposed method compared to 82.2% for standard ICL with a 540B Flan-PaLM model.

| Method | Model Size | | |
|---|---|---|---|
| | 2B | 9B | 27B |
| *FT-Only* | 77.7% | 81.2% | 82.6% |
| *ICL-Only* | 74.9% | 82.1% | 82.5% |
| *ICL+FT* | **84.3%** | **85.1%** | **86.4%** |

*FT-Only*. *ICL-Only* on the other hand is generally not able to make effective use of additional in-context examples. Figure 3 provides a holistic view of performance scaling with respect to data availability and model size. It shows the average performance across all BBH tasks, varying either the number of training examples for a fixed model size or as a function of model size for a fixed training data budget. We summarize the results on BBH in Table 1 and highlight that over a wide range of data-budgets and model-sizes *ICL+FT* surpasses models that are three times larger; or, alternatively for same-sized models reduces the data-demand by a factor of 3-10 to obtain the same test-set performance.

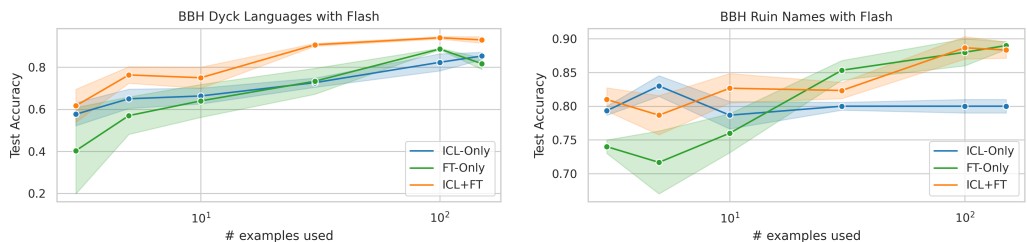

Figure 2: Two representative results from the Big-Bench-Hard suite with between 3 and 150 ground truth examples. We observe the typical signature where *ICL+FT* performs on-par with *ICL-Only* when very few examples are available, but improves similarly or better than *FT-Only* with more training data. Appendix C.1 shows the individual results on all 23 BBH tasks.

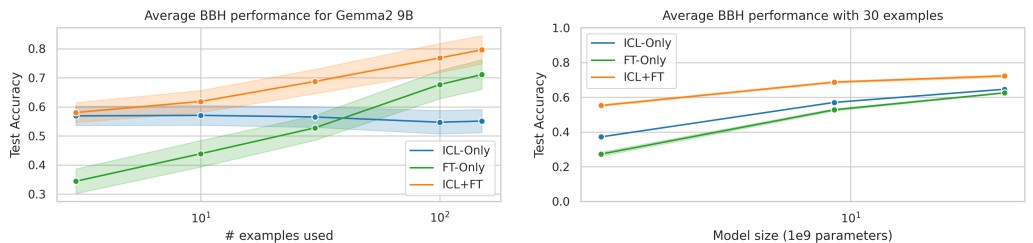

Figure 3: Average test-set performance across 23 BBH tasks. **Left**: Performance as a function of the number of ground-truth examples (3 to 150) for a fixed model size. **Right**: Performance as a function of model size (2B to 27B parameters) for a fixed number of training examples.

**NLP task suite.** We compare the methods on the 11 evaluation tasks used by Wei et al. (2023). Many of these have only 50, the rest up to 100 examples available. We use the same hyperparameter search-space as before. To estimate held-out performance for the prequentially trained models we repeat training on 5 different permutations of the data and report the average accuracy of the last 10 examples for the best hyper-parameter configuration according to the prequential average accuracy. Thus effectively reporting the expected performance estimated on 50 held-out examples. For *ICL-Only* we use 1, 2, 5, 10, 15 and 20 in-context examples and report the best-performing choice. We observe that *ICL-Only* scales better with number of examples than on the BBH datasets and performance often increases up to 10-20 in-context examples before saturating. Without exception, *ICL+FT* is on-par or exceeding *ICL-Only* for all model sizes. Furthermore, on many tasks the 9B or even the 2B model remains competitive or outperforms 27B *ICL-Only*, leading to significant compute and energy savings if such a model was deployed. Especially considering that the provided context size per inference is usually less than half of *ICL-Only* (Table 2 and Appendix C.2).

**Parity-20 Task.** To demonstrate the in-context capabilities of LLMs Agarwal et al. (2024) propose a task where the model is presented with 20 digit long `0` or `1` strings and is expected to compute the parity by producing `Even` or `Odd` for each digit. We confirm the positive correlation between the number of in-context examples and performance for Gemini Pro 1.5 *ICL-Only* and observe a sequence match accuracy of about 20% when providing 100 in-context examples, which is consistent with their findings. Gemma-2 27B *ICL-Only* performs significantly weaker, struggling to reach 2% accuracy even with 300 examples. However, fine-tuning leads to dramatic improvements: *FT-Only* requires about 100 examples to achieve 100% accuracy, and *ICL+FT* requires only about 30 examples (Fig. 4, left). In conclusion: While *ICL-Only* shows consistent improvements with more in-context examples, it's overall sample-efficiency is more than an order of magnitude weaker than fine-tuning based approaches.

**FLoRes: Low-resource language translation.** Building upon previous research that explored the advantages of *ICL-Only* for translating English into low-resource languages (Agarwal et al., 2024), we expand our investigation to encompass both *ICL+FT* and *FT-Only*. Figure 4 shows the results for translating from English into Kurdish and Bemba, utilizing the FLoRes-200 benchmark dataset,

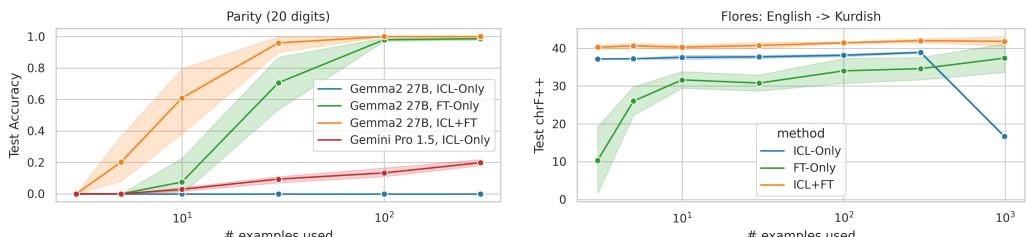

Figure 4: **Left**: Parity-20 task from Agarwal et al. (2024) **Right**: FLoRes English to Kurdish translation task with Gemma-2 27B.

Table 3: Average BBH test-set accuracy when using (per-dataset) prequential hyper-parameter selection (*per ds* column) vs. using fixed, globally chosen hyper-parameters (*global* column). We consider the scenarios with either 30 or 150 ground-truth examples. All accuracies have $\pm 2\%$ confidence estimated over 5 seeds. *ICL+FT* is more robust to *hp* choices and degrades only minimally when forgoing individual *hp* selection completely. See Appendix C.10 for details.

|  |  | 30 examples | | | 150 examples | | |
|---|---|---|---|---|---|---|---|
|  |  | per ds | global | $\Delta$ | per ds | global | $\Delta$ |
| *FT-Only* | 27B | 62.6% | 60.3% | 2.2% | 72.6% | 66.7% | 5.9% |
|  | 9B | 52.8% | 49.8% | 2.9% | 69.7% | 60.2% | 9.5% |
|  | 2B | 27.3% | 25.6% | 1.7% | 50.7% | 46.3% | 4.4% |
| *ICL+FT* | 27B | 72.3% | 72.6% | -0.3% | 81.8% | 79.1% | 2.7% |
|  | 9B | 68.8% | 69.1% | -0.3% | 78.7% | 77.7% | 1.0% |
|  | 2B | 55.3% | 54.9% | 0.4% | 67.5% | 65.4% | 2.1% |

which comprises 1000 sentence pairs (NLLB Team, 2022). We observe that *ICL+FT* yields marginal yet consistent improvements over *ICL-Only*, though additional improvements from more translation data become barely discernible. In the inverse translation direction, from low-resource languages into English, both fine-tuning methods led to a slight, yet noticeable, performance decrease compared to *ICL-Only* (see Appendix C.6). We hypothesize that the models primarily leverage translation skills acquired during pre-training, and the limited number of examples provided here may not suffice for substantial skill enhancement. For translation into English, the typically well-trained model might lose some refinement through gradient updates.

**Prequential hyperparameter selection.** There are two aspects related to hyperparameter selection that we investigate here: a) Performance with global hyperparameters: How do the approaches perform when using a single set of hyperparameters across all datasets, rather than tuning them individually? b) Comparison to standard fine-tuning: Is prequential training and hyperparameter selection competitive with standard i.i.d. fine-tuning, where training batches are freely sampled from a training set?

To investigate the first aspect, we select the single best-performing hyperparameter configuration per method and model size, based on average test-set performance. It's crucial to emphasize that this selection process utilizes extensive held-out data, which might not be available in practical scenarios. Table 5 shows the average results over all BBH tasks when either using this fixed hyperparameter configuration, or the prequential selection process. We observe two consistent trends: Automatic selection becomes more impactful when the size of the available data grows; and that *ICL+FT* is consistently more robust and maintains good performance even with a single globally chosen hyper-parameter configuration.

To assess the effectiveness of prequential training compared to standard i.i.d. training, we conducted a comprehensive hyperparameter sweep with i.i.d. training, selecting the hyperparameters that yielded the best overall performance across all 23 BBH test sets. Again, it is crucial to acknowledge that this procedure relies on a substantial amount of held-out data for hyperparameter selection. Table 6 shows that for *FT-Only* , the i.i.d. approach achieves slightly better average results than prequential training. This aligns with observations in the vision domain (Bornschein et al.,

Table 4: Prequential training (Alg. 1) vs. standard i.i.d. training with Gemma-2 9B. We show the averaged accuracy over 23 BBH test-sets. Note that globally-chosen hyper-parameters introduce information leakage as a large number of test-set examples are used to chose these. All accuracies have $\pm 2\%$ confidence estimated over 5 seeds. We observe that globally chosen hyper-parameters with iid. training are competitive with prequential per-dataset tuning; however iid. training does not provide any reliable performance metric without additional held-out data.

| Training | 30 examples | | | 150 examples | | |
| Hyper-params | preq per ds | preq global | iid global | preq per ds | preq global | iid global |
|---|---|---|---|---|---|---|
| *FT-Only* | 52.8% | 49.8% | 53.2% | 69.7% | 60.2% | 68.1% |
| *ICL+FT* | 68.8% | 69.1% | 68.9% | 78.7% | 77.7% | 78.7% |

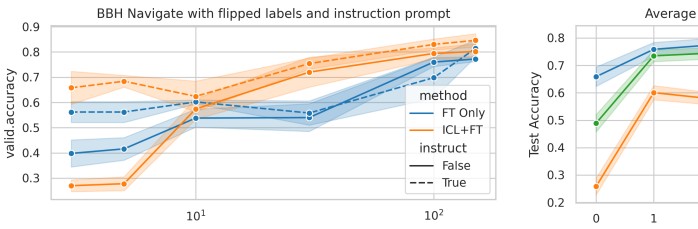

Figure 5: **Left**: We revisit the BBH Navigate task with flipped labels (Fig. 15), however optionally provide an instruction about the flipped labels in the prompt. **Right**: Varying the number of in-context examples used during training and testing, averaged over a subset of the BBH tasks. The left most point corresponds to *FT-Only* .

2023), where i.i.d. training of neural networks can yield marginally better performance when disregarding the additional data needed for hyperparameter tuning. However, this advantage largely disappears when leveraging the efficient hyperparameter selection enabled by prequential training, which eliminates the need for separate validation data. Conversely, for *ICL+FT* , all training and hyperparameter tuning approaches resulted in essentially the same test-set performance. This reinforces our observation that *ICL+FT* is a robust and well-behaved approach requiring less tuning than *FT-Only* .

**Interaction with prompt-tuning.** Thus far, our experiments have relied solely on in-context examples without any explicit task instructions within the prompt. However, prompt engineering —the art of crafting verbal cues to guide language models toward desired behavior— is a well-established practice when using LLMs. We here investigate how prefixing the fine-tuning data with an instructional prompt can enhance the performance of *FT-Only* and *ICL+FT* approaches. Figure 5 shows the results an the BBH Navigate task with flipped labels; this time however with optionally prefixing each finetuning example with *"Important: All answers are flipped! Yes is No and No is Yes! Always reply with the opposite of what you consider the right answer!"*. We provide the same prompt at test time when performing predictions and observe that providing such instruction leads to a significant uplift in performance for both methods.

**Number of in-context examples.** We investigate the impact of varying the number of in-context examples used during training and inference. Due to computational constraints, we conducted this experiment on a subset of eight BBH tasks. Figure 5 (right) shows the average results across these datasets. As detailed in Appendix C.3, we observed that increasing the number of in-context examples from zero to one yields the most significant performance gain. Further increases up to around five examples provide marginal improvements, after which performance plateaus or even declines.

We also explored varying the number of in-context examples independently for training and inference. However, these experiments revealed significant variability across datasets and training set sizes, with no consistently discernible patterns. Appendix C.3 presents a selection of these results.

Table 5: Average BBH test-set accuracy when using (per-dataset) prequential hyper-parameter selection (*per ds* column) vs. using fixed, globally chosen hyper-parameters (*global* column). We consider the scenarios with either 30 or 150 ground-truth examples. All accuracies have $\pm 2\%$ confidence estimated over 5 seeds. *ICL+FT* is more robust to *hp* choices and degrades only minimally when forgoing individual *hp* selection completely. See Appendix C.10 for details.

| | | 30 examples | | | 150 examples | | |
|---|---|---|---|---|---|---|---|
| | | per ds | global | $\Delta$ | per ds | global | $\Delta$ |
| *FT-Only* | 27B | 62.6% | 60.3% | 2.2% | 72.6% | 66.7% | 5.9% |
| | 9B | 52.8% | 49.8% | 2.9% | 69.7% | 60.2% | 9.5% |
| | 2B | 27.3% | 25.6% | 1.7% | 50.7% | 46.3% | 4.4% |
| *ICL+FT* | 27B | 72.3% | 72.6% | -0.3% | 81.8% | 79.1% | 2.7% |
| | 9B | 68.8% | 69.1% | -0.3% | 78.7% | 77.7% | 1.0% |
| | 2B | 55.3% | 54.9% | 0.4% | 67.5% | 65.4% | 2.1% |

Table 6: Prequential training (Alg. 1) vs. standard i.i.d. training with Gemma-2 9B. We show the averaged accuracy over 23 BBH test-sets. Note that globally-chosen hyper-parameters introduce information leakage as a large number of test-set examples are used to chose these. All accuracies have $\pm 2\%$ confidence estimated over 5 seeds. We observe that globally chosen hyper-parameters with iid. training are competitive with prequential per-dataset tuning; however iid. training does not provide any reliable performance metric without additional held-out data.

| | 30 examples | | | 150 examples | | |
|---|---|---|---|---|---|---|
| Training Hyper-params | preq per ds | preq global | iid global | preq per ds | preq global | iid global |
| *FT-Only* | 52.8% | 49.8% | 53.2% | 69.7% | 60.2% | 68.1% |
| *ICL+FT* | 68.8% | 69.1% | 68.9% | 78.7% | 77.7% | 78.7% |

**LoRA fine-tuning.** We explore using LoRA (Hu et al., 2021) as a memory efficient alternative to full fine-tuning, repeating the complete suite of BBH experiments with Gemma-2 9B. Averaged over the 23 BBH datasets, LoRA fine-tuning slightly improved *FT-Only* performance, increasing from 52.8% to 59.1% with 30 examples and from 69.7% to 70.2% with 150 examples. For *ICL+FT* the results remained largely unchanged, with performance changing from 68.8% (full fine-tuning) to 67.6% (LoRA) with 30 examples, and from 78.7% to 78.9% with 150 examples. These results, with further details in Appendix C.8, suggest that LoRA fine-tuning has minimal impact on the predictive performance of of either *FT-Only* and *ICL+FT* . All previously observed trends remain qualitatively unchanged for our benchmark data-sets.

## 5 CONCLUSIONS

In this study, we systematically investigated how to effectively adapt LLMs to downstream tasks when ground-truth data is limited. We rigorously compared in-context learning with a fixed model (*ICL-Only*), traditional fine-tuning (*FT-Only*), and a unified approach that involves fine-tuning on k-shot in-context examples (*ICL+FT*). Our empirical study demonstrated that *ICL+FT* consistently matches or exceeds the performance of both baselines, especially in data-scarce scenarios. The success of *ICL+FT* highlights its ability to leverage the complementary strengths of both approaches by incorporating task-specific examples directly into the fine-tuning process, enabling the model to learn more efficiently.

Recognizing the challenges of hyperparameter tuning in the small-data regime, we introduced a practical and efficient protocol based on prequential evaluation. This method eliminates the need for a separate validation set, and our experiments demonstrated its robustness. The combined strategy provides a readily applicable solution for adapting LLMs to new tasks, particularly when data is scarce.

## REPRODUCIBILITY

Our core methodology, including the ICL+FT approach and prequential hyperparameter selection, is formally described in Section 3 and detailed in Algorithm 1. All experiments were conducted using publicly available Gemma-2 models and established benchmarks (e.g., BBH, FLoRes-200). Our gradient-based fine-tuning implementation is based on the official open-source Gemma codebase with only minor changes to the training loop to accommodate the prequential procedure. Beginning of Section 4 outlines the general experimental setup, including all details regarding optimizer, hyperparameter search space, and seeding strategy. Appendix B provides the concrete prompt templates (B.1) and the precise logic for response parsing (B.2).

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

# SUPPLEMENTARY MATERIALS

## A    PREQUENTIAL AVERAGE PERFORMANCE

This prequential metric assesses predictive performance during sequential data ingestion, eliminating the need for separate test sets and balancing the influence of small and large data regimes. Grounded in the principles of prequential analysis (Dawid, 1984; Philip Dawid and Vovk, 1999) and prequential Minimum Description Length (MDL) (Grünwald, 2007; Grünwald and de Rooij, 2005), it has been thoroughly studied and is closely related to marginal likelihood-based model selection(Lyle et al., 2020) and leave-k-out cross-validation (Fong and Holmes, 2020). Recent applications in deep learning include facilitating causal structure discovery (Bornschein et al., 2021), performing neural architecture search (Ru et al., 2021), and evaluating visual representations (Li et al., 2023). In this work, we specifically leverage prequential evaluation for its effectiveness in hyperparameter selection, particularly in the small-data regime. However, it can be valuable metric for benchmarking LLM adaptation methods, regardless of the specific technique employed (in-context learning, fine-tuning, retrieval augmentation, etc.). We therefore also report the obtained prequential performance for all our main experiments in Appendix A.2.

### A.1    HYPERPARAMETER SELECTION WITH A PREQUENTIAL METRIC

To illustrate prequential hyperparameter selection in practice, we present results from experiments on BBH datasets with 150 examples. Using a fixed random data order, we consider four hyperparameter configurations with learning rates of {1e-4, 3e-4} and training epochs of {5, 10}. The left side of Figure displays the average accuracy ($L_i$) as a function of examples seen ('i'), while the right side shows the average test-set performance on 100 additional examples from the same distribution.

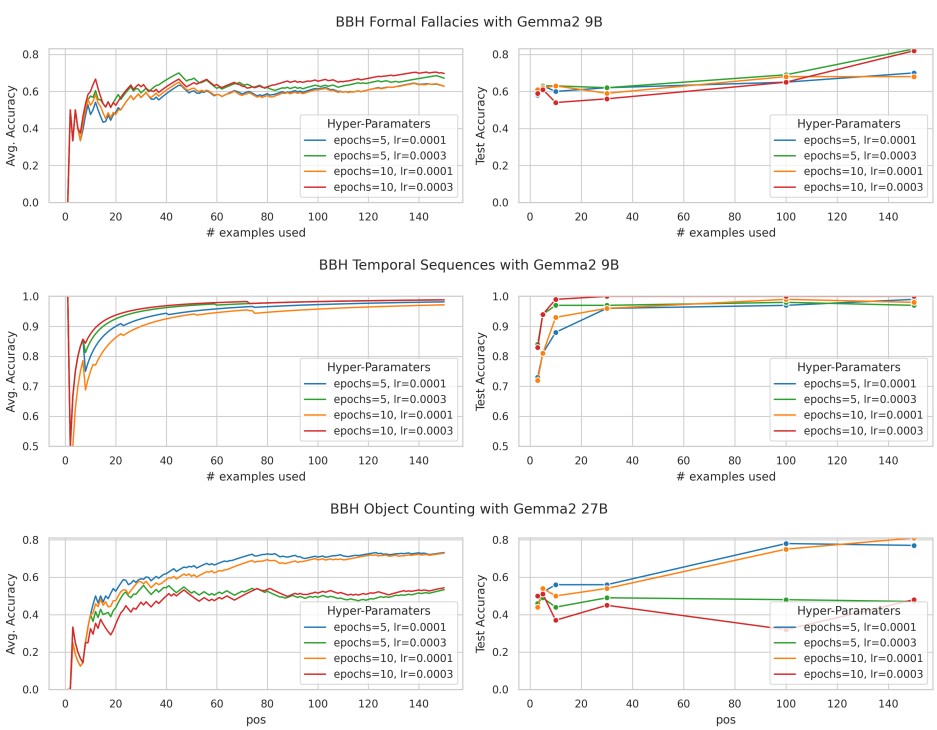

Figure 6: **Left** Prequential avg. accuracy. **Right** Corresponding test-set accuracies after 3, 5, 10, 30, 100, and 150 training examples.

Generating the right-hand side plot required not only 100 extra ground-truth examples but also significantly more computation. Sampling from a transformer is a sequential operation, often much

slower than computing log-losses or gradients. Each data point on the right represents 100 sampling operations, resulting in a total of 600 continuations for each hyperparameter configuration. In contrast, the left-hand side curve required only 150 sampling operations (one per training data point). For our implementation, test-set evaluation consumed approximately half to two-thirds of the total runtime in our experiments. Fully adopting the prequential approach would allow us to utilize all available data for training and significantly reduce runtimes by avoiding costly held-out evaluations.

As established in prior work (Philip Dawid and Vovk, 1999; **?**), the prequential metric (left) converges to a lower bound on the expected held-out performance for i.i.d. data.

## A.2 PREQUENTIAL AVERAGE ACCURACY

To facilitate comparison in future research, we here report prequential accuracies for all our main experiments.

Table 7: Prequential average accuracy for the 23 BBH tasks form Section 4 and Appendix C.1.

| | Gemma-2 2B | | Gemma-2 9B | | Gemma-2 27B | |
|---|---|---|---|---|---|---|
| | *FT-Only* | *ICL+FT* | *FT-Only* | *ICL+FT* | *FT-Only* | *ICL+FT* |
| Boolean Expressions | 46.1% | 82.1% | 58.2% | 88.2% | 82.8% | 92.9% |
| Causal Judgement | 26.0% | 61.7% | 54.0% | 67.1% | 60.2% | 70.6% |
| Date Understanding | 72.4% | 72.9% | 78.3% | 79.0% | 79.6% | 80.6% |
| Disambiguation Question Answering | 43.8% | 78.5% | 80.7% | 85.7% | 82.7% | 92.6% |
| Dyck Languages | 49.0% | 79.2% | 64.8% | 87.7% | 82.0% | 91.0% |
| Formal Fallacies | 49.5% | 52.8% | 61.2% | 68.4% | 60.9% | 74.1% |
| Geometric Shapes | 41.2% | 75.8% | 58.3% | 84.7% | 66.9% | 84.1% |
| Hyperbaton | 1.4% | 85.2% | 82.9% | 94.7% | 90.0% | 94.5% |
| Logical Deduction (5 Objects) | 33.9% | 39.1% | 35.2% | 62.7% | 58.9% | 69.8% |
| Movie Recommendation | 29.3% | 79.6% | 68.9% | 95.5% | 93.3% | 96.2% |
| Multi-Step Arithmetic (Two Steps) | 1.9% | 2.7% | 2.8% | 4.4% | 4.0% | 5.1% |
| Navigate | 41.2% | 65.1% | 65.1% | 77.1% | 65.3% | 84.8% |
| Object Counting | 32.0% | 55.0% | 57.7% | 69.1% | 69.6% | 74.4% |
| Penguins in a Table | 25.2% | 42.4% | 60.4% | 65.9% | 57.2% | 63.3% |
| Reasoning about Colored Objects | 45.7% | 59.4% | 74.9% | 82.5% | 80.2% | 87.5% |
| Ruin Names | 30.4% | 78.9% | 86.7% | 89.6% | 83.3% | 88.6% |
| Salient Translation Error Detection | 35.4% | 42.1% | 64.9% | 65.1% | 60.0% | 69.5% |
| Snarks (Graph Theory) | 56.9% | 71.2% | 50.3% | 80.1% | 52.1% | 83.1% |
| Sports Understanding | 49.2% | 78.8% | 85.1% | 90.6% | 86.2% | 92.9% |
| Temporal Sequences | 36.1% | 91.6% | 90.9% | 98.8% | 99.7% | 99.2% |
| Tracking Shuffled Objects (5 Objects) | 7.7% | 23.1% | 19.7% | 34.9% | 28.9% | 39.3% |
| Web of Lies | 37.9% | 56.9% | 57.7% | 92.1% | 67.3% | 91.1% |
| Word Sorting | 18.3% | 23.2% | 49.1% | 49.0% | 62.7% | 65.8% |
| **Avg. All** | **35.2%** | **60.7%** | **61.2%** | **74.5%** | **68.4%** | **77.9%** |

Table 8: Prequential average accuracy for the 11 NLP task form Section 4 and Appendix C.2.

| | Gemma-2 2B | | Gemma-2 9B | | Gemma-2 27B | |
|---|---|---|---|---|---|---|
| | *FT-Only* | *ICL+FT* | *FT-Only* | *ICL+FT* | *FT-Only* | *ICL+FT* |
| ADEC | 74.8% | 84.4% | 77.6% | 87.5% | 83.2% | 87.2% |
| OR | 84.8% | 93.9% | 94.8% | 97.6% | 92.0% | 96.0% |
| SOT | 85.8% | 91.9% | 92.7% | 94.0% | 94.6% | 95.7% |
| SUBJ | 80.4% | 92.2% | 84.2% | 94.1% | 87.4% | 95.4% |
| TC | 82.4% | 91.1% | 85.6% | 91.0% | 90.0% | 91.9% |
| TEAB | 66.7% | 74.4% | 71.5% | 79.2% | 70.6% | 80.5% |
| TEAT | 56.9% | 68.3% | 63.5% | 72.7% | 63.5% | 73.1% |
| TEFE | 60.3% | 67.0% | 66.0% | 73.4% | 69.6% | 74.4% |
| TEH | 60.4% | 71.2% | 65.4% | 76.1% | 66.8% | 77.2% |
| TEHI | 57.1% | 66.0% | 63.2% | 68.4% | 64.6% | 70.0% |
| TOS | 70.8% | 82.8% | 74.4% | 83.8% | 76.4% | 84.2% |
| **Avg. All** | **70.9%** | **80.3%** | **76.3%** | **83.4%** | **78.1%** | **84.1%** |

# B  EXPERIMENTAL DETAILS

## B.1  PROMPT TEMPLATE

We show an example training sequence from the BBH Navigate task. During training, we place a loss on all desired replies from the model – here marked by underlining. These tokens correspond to the $y_i$ from the context examples, and the $y$ from the final target example. All other tokens are considered context and can be causally attended to, but do not directly contribute gradients.

```
If you follow these instructions, do you return to the starting point?
Always face forward. Take 10 steps forward. Take 8 steps left. Take 3
    steps backward. Take 7 steps right. Take 2 steps forward. Take 4
    steps forward. Take 10 steps forward.
Options:
- Yes
- No
Answer: No

== Next Example ==
If you follow these instructions, do you return to the starting point?
Take 7 steps. Take 8 steps. Take 10 steps. Turn around. Turn around.
    Take 5 steps. Turn around.
Options:
- Yes
- No
Answer: No

== Next Example ==
If you follow these instructions, do you return to the starting point?
Turn left. Turn right. Take 8 steps. Take 4 steps. Turn right. Turn
    right. Take 10 steps. Take 1 step. Take 1 step.
Options:
- Yes
- No
Answer: Yes
```

## B.2  RESPONSE PARSING

To account for potential idiosyncrasies introduced by instruction-tuned models, we apply the following regular expressions when comparing the generated response to the ground-truth label. The first matching expression is used to extract the answer for verbatim comparison: ″r/Answer: (.+)/″, ″r/The answer is (.+)/″, ″r/**(.+)**/″, ″r/(.+)/″

# C    ADDITIONAL RESULTS

## C.1    BIG-BENCH HARD RESULTS

All experiments are run with the standard hyperparameter cube and 5 different seeds to obtain $2\sigma$ uncertainty intervals.

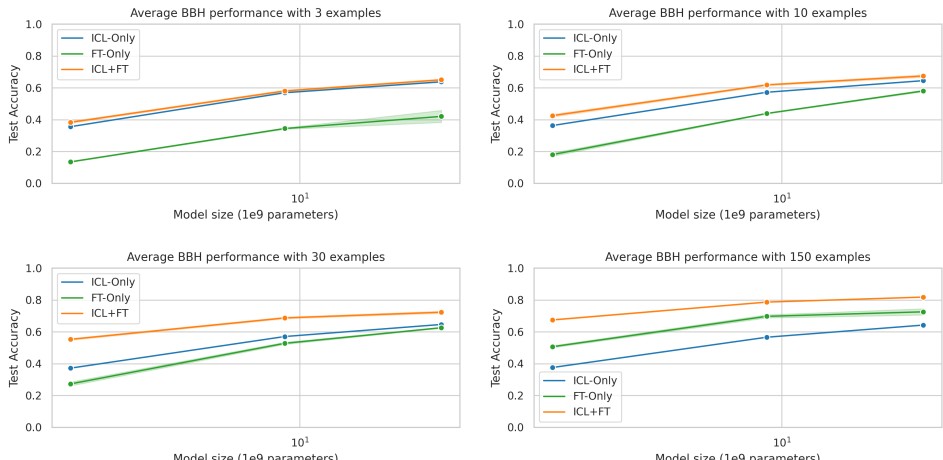

Figure 7: Average BBH test-set performance for different methods and model-sizes: **Top Left:** with 3, **Top Right:** with 10, **Bottom Left:** with 30 ground-truth examples available; Gemini 1.5 Pro *ICL-Only* achieves **72.8%**. **Bottom Right:** with all but 100 ground-truth examples available (= 150 training examples for most datasets). Gemini 1.5 Pro *ICL-Only* achieves **76.6%**

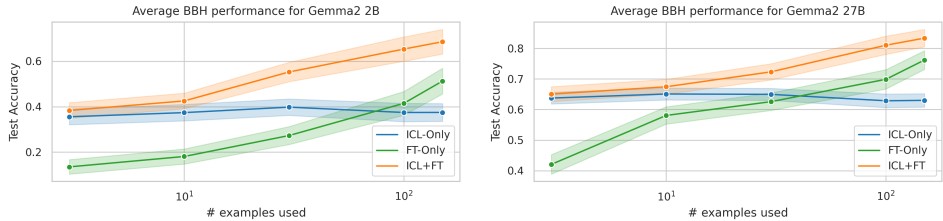

Figure 8: Average BBH test-set performance for different number of ground-truth examples: **Left:** For Gemma-2 2B; **Right:** Gemma-2 27B. Figure 3 in the main paper shows the results for Gemma-2 9B

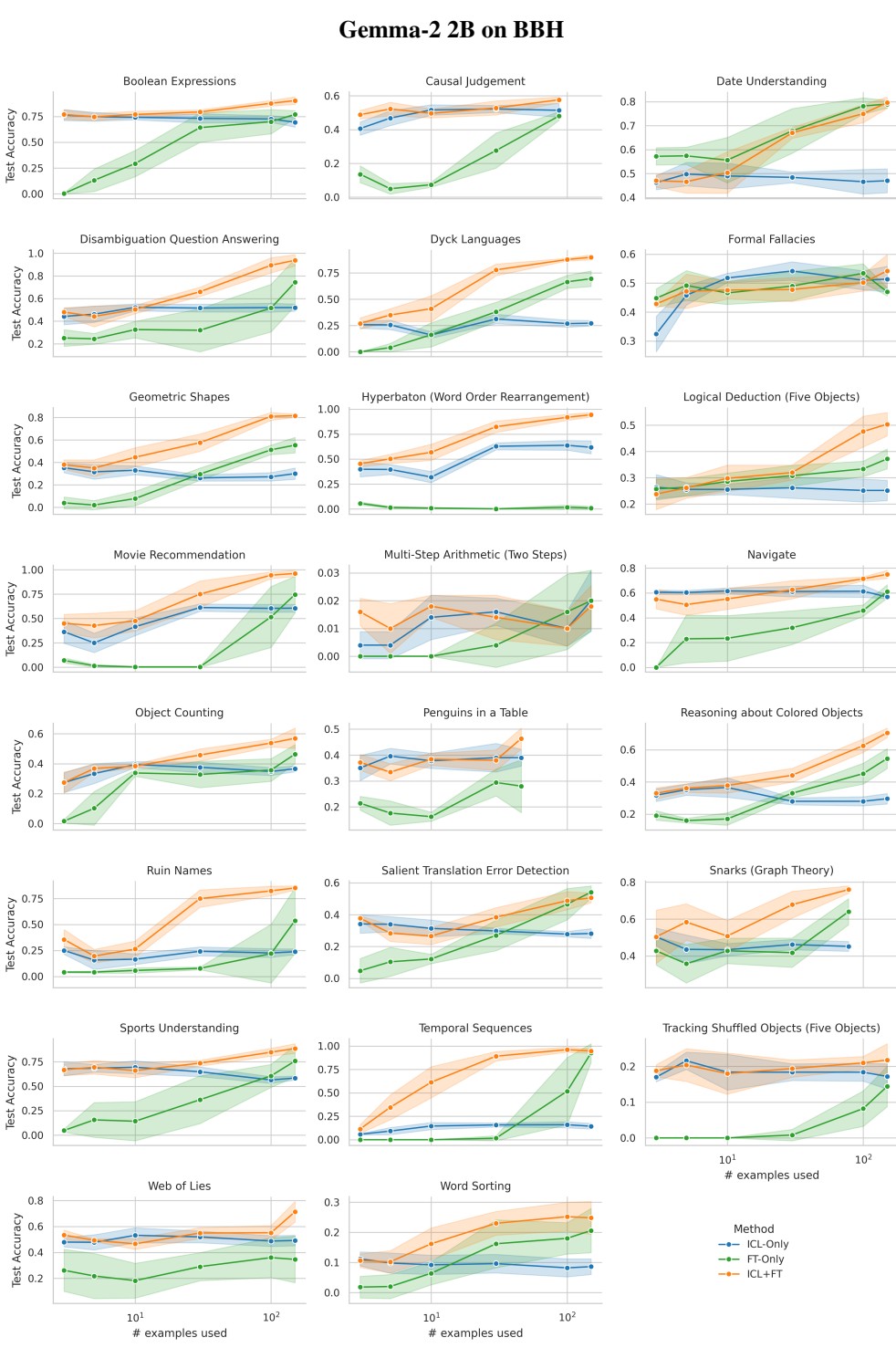

Figure 9: Full suite of results for Gemma-2 2B on all 23 individual BBH datasets. All experiments are run with the standard hyperparameter cube and 5 different seeds to obtain $2\sigma$ uncertainty intervals.

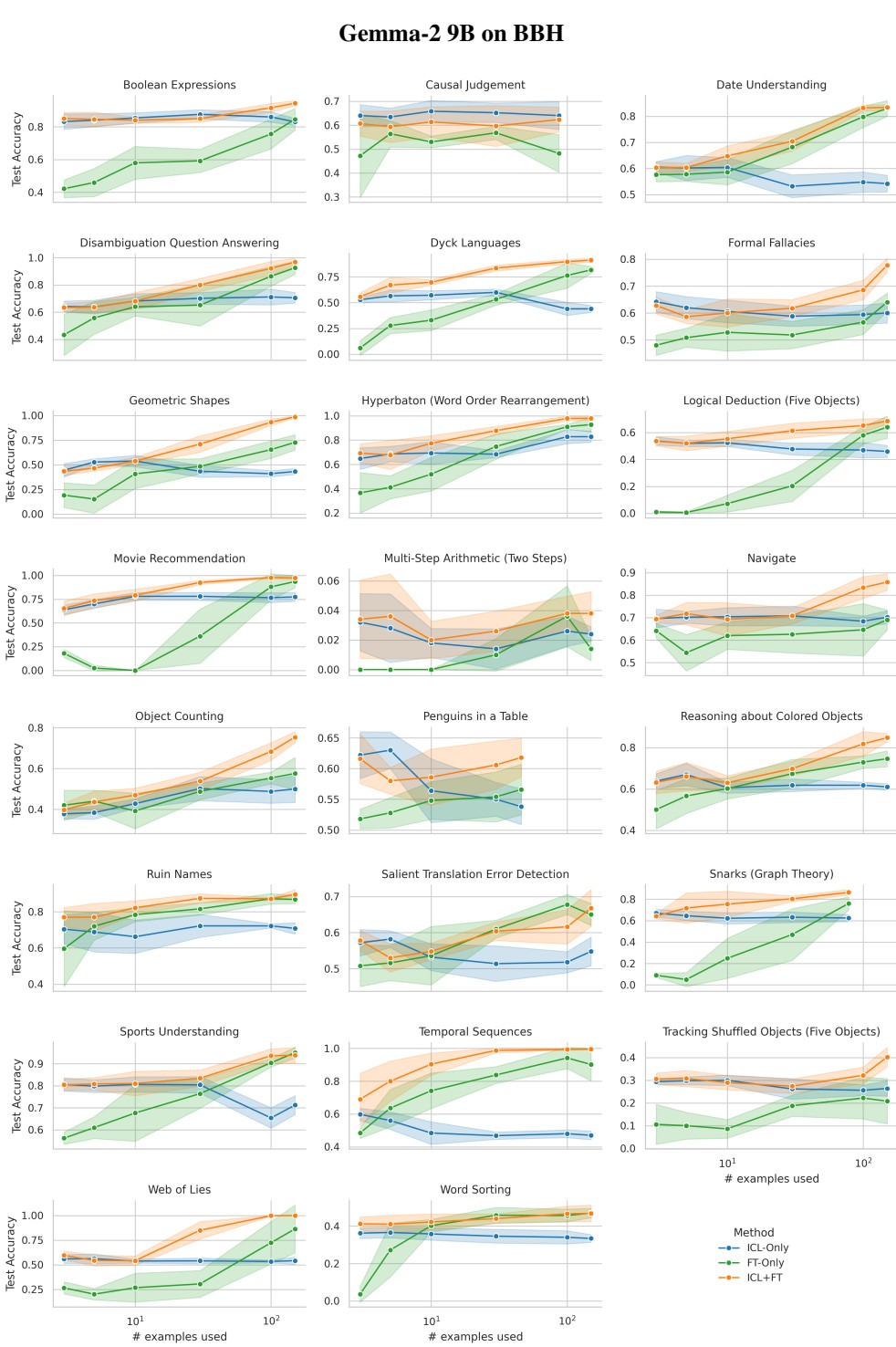

Figure 10: Full suite of results for Gemma-2 9B on all 23 individual BBH datasets. All experiments are run with the standard hyperparameter cube and 5 different seeds to obtain $2\sigma$ uncertainty intervals.

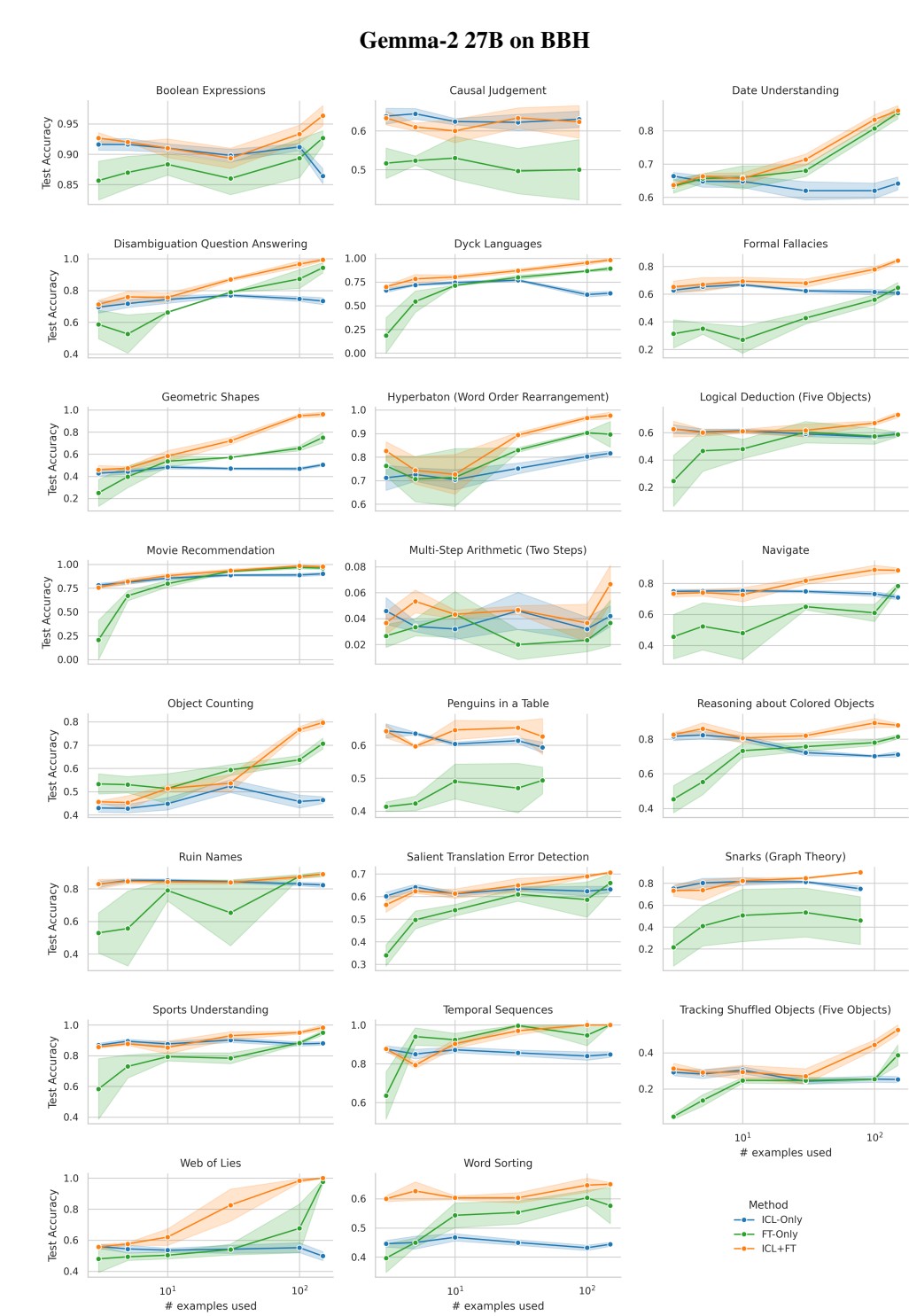

Figure 11: Full suite of results for Gemma-2 27B on all 23 individual BBH datasets. All experiments are run with the standard hyperparameter cube and 5 different seeds to obtain $2\sigma$ uncertainty intervals.

## C.2  11 TASKS FROM THE SYMBOL-TUNING PAPER

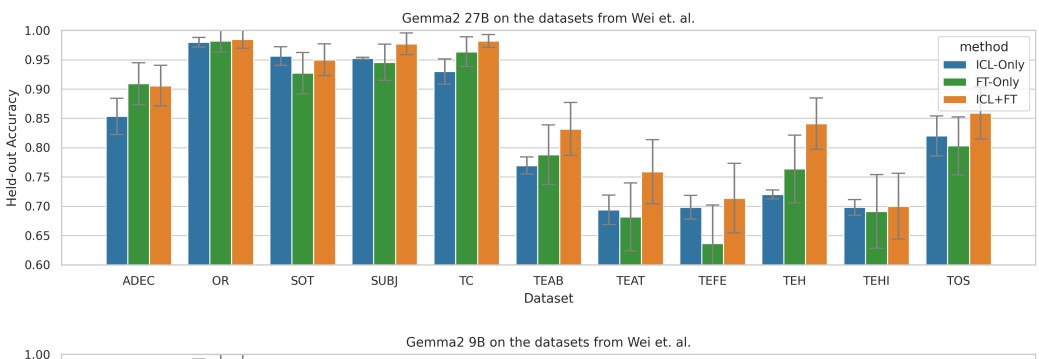

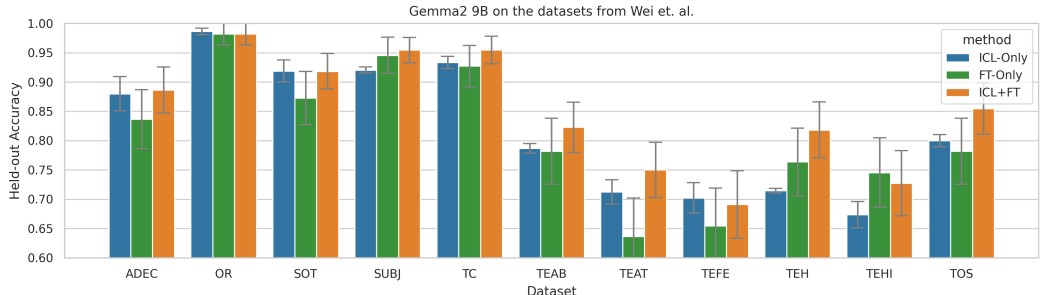

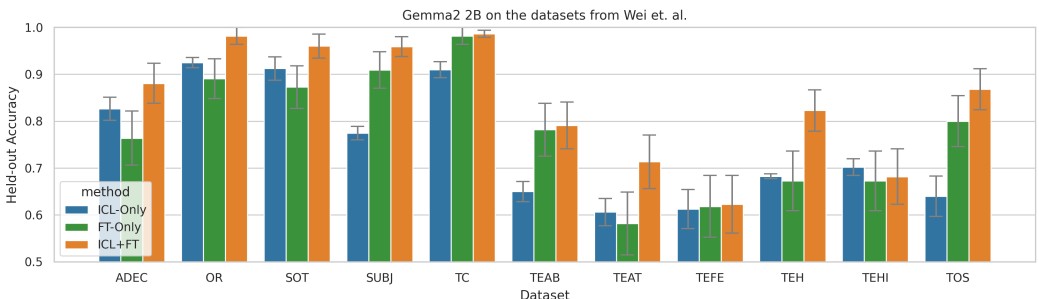

Figure 12: Individual results for the 11 evaluation data sets used in Section 4:

## C.3 NUMBER OF IN-CONTEXT EXAMPLES

Varying the number of in-context examples used for training and testing with *ICL+FT* . We here select a subset of 8 representative datasets from BBH.

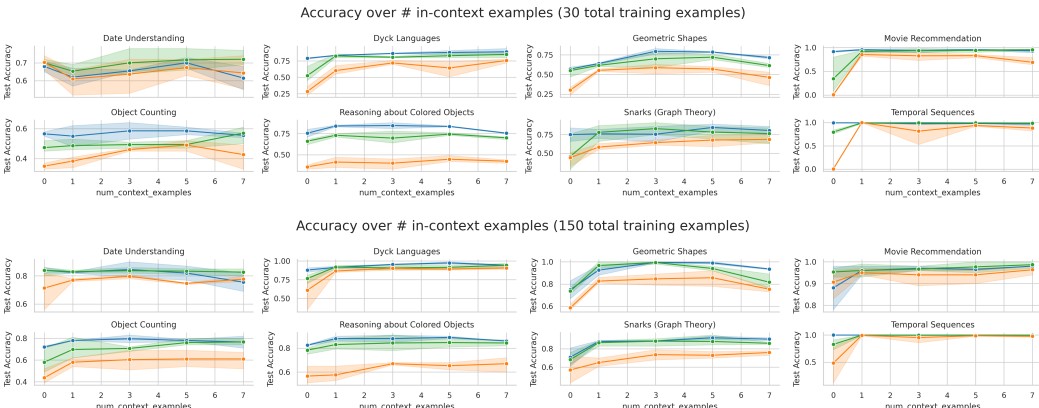

Figure 13: Test-set performance when varying the number of in-context examples for individual BBH datasets with *ICL+FT* . The left most points at 0 correspond to *FT-Only* . We use the same number of in-comntext examples at training and test-time.

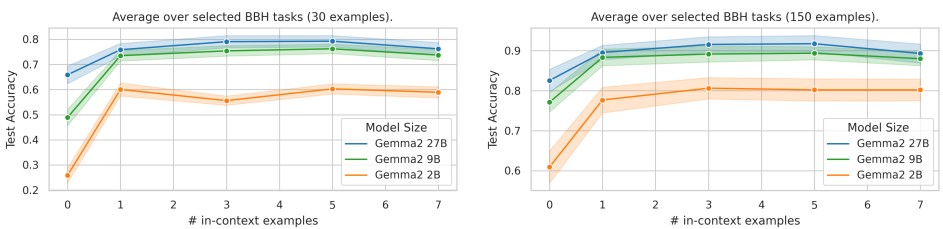

Figure 14: *ICL+FT* test-set performance averaged over the 8 BBH data-sets.

## C.4 FLIPPED LABELS

Previous work has explored the effectiveness of in-context learning when downstream tasks use labels that deviate from the model's pretraining biases (Kossen et al., 2023) . We examine the BBH Navigate task, where the model predicts whether a sequence of steps leads back to the starting point. We compare the methods on the original task and a version with flipped labels where the model has to reply *No* when one would return, and *Yes* when not. Note that *FT-Only* is closer to random performance when using less then 10 examples, which gives it seemingly an advantage over the ICL based methods, that are solving the logical challenge, but require examples to learn about the label-inversion (Fig. 15). As reported previously (Kossen et al., 2023), *ICL-Only* struggles severely with flipped labels, while fine-tuning approaches and especially *ICL+FT* perform significantly better.

## C.5 DETAILED LOOK AT NEXT-STEP PERFORMANCE

Previously, we assessed sample efficiency by reporting average test-set performance after training on various-sized subsets of the data. However, to visualize the generalization performance we can directly plot the next-example loss during sequential data ingestion, as these are guaranteed to be unseen examples. To obtain the expected performance after observing $i$ datapoints, we average over multiple data permutations. We demonstrate this by performing 500 runs with different permutations for a fixed hyper-parameter configuration on Big-Bench-Hard Navigate and Geometric Shapes tasks. These plots reveal the generalization performance after encountering exactly $i$ examples.

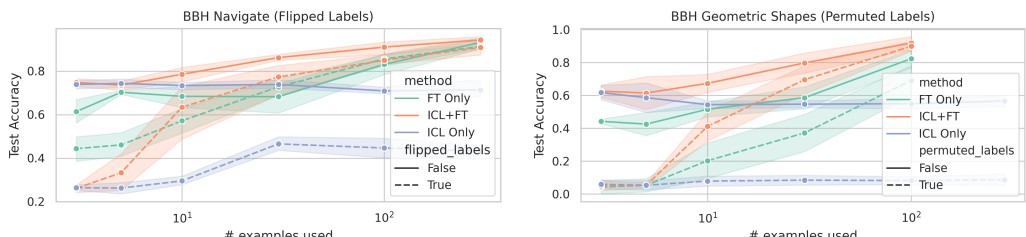

Figure 15: **Left**: Gemma-2 27B on the BBH Navigate binary classification task with standard, and flipped labels. **Right**: BBH Geometric-shapes is a 5-way classification task. Note that for the permuted tasks, all methods initially perform worse than a random predictor because their pretraining biases are violated. See Appendix B for example prompts.

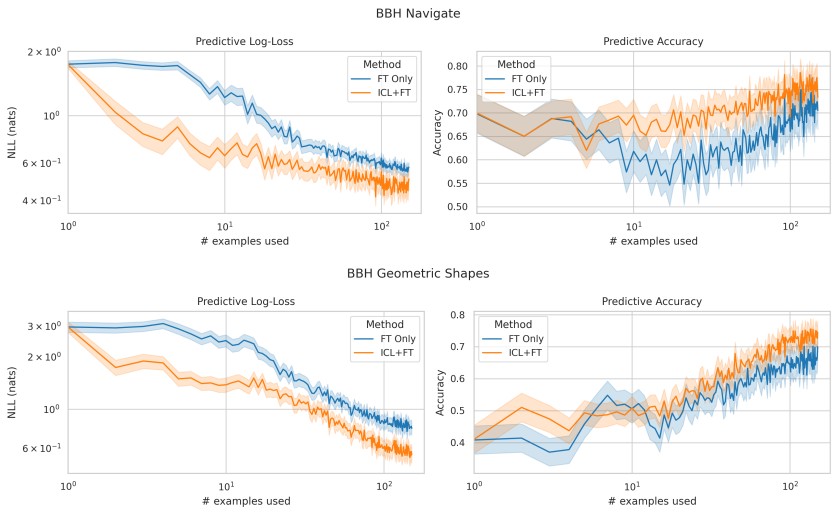

Figure 16: Next-step generalization NLL and accuracy as a function of data seen so far (averaged over 500 random data-permutations). We finetune a Gemma-2 9B model with a single fixed hyper-parameter configuration (num-epochs=1, learning-rate=2e-4).

### C.6    FLORES LOW-RESOURCE MACHINE TRANSLATION

We evaluate our approach on English translation tasks involving two low-resource languages from the FLORES data set (NLLB Team, 2022): Kurdish and Bemba. In addition to the English-to-target-language direction, we here also report results for the target-language-to-English translation, a direction that is typically not evaluated in prior work on in-context learning for machine translation. Here, for the first time, we observe that any kind of finetuning (*FT-Only* and *ICL+FT* ) have a slight, but statistically negative effect on the performance. We hypothesize that the models primarily leverage translation skills acquired during pre-training, and the limited number of examples provided here may not suffice for substantial skill enhancement. For translation into English, the typically well-trained modelmight lose some refinement through gradient updates.

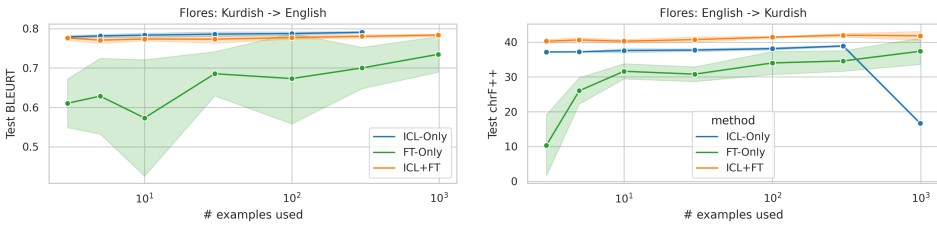

1242
1243
1244
1245
1246
1247
1248
1249
1250
1251
1252
1253
1254
1255
1256
1257
1258
1259
1260

## C.7 DATASETS FROM LONG-CONTEXT ICL PAPER

We run experiments with Gemma-2 9B on the datasets used by Bertsch et al. (2024).

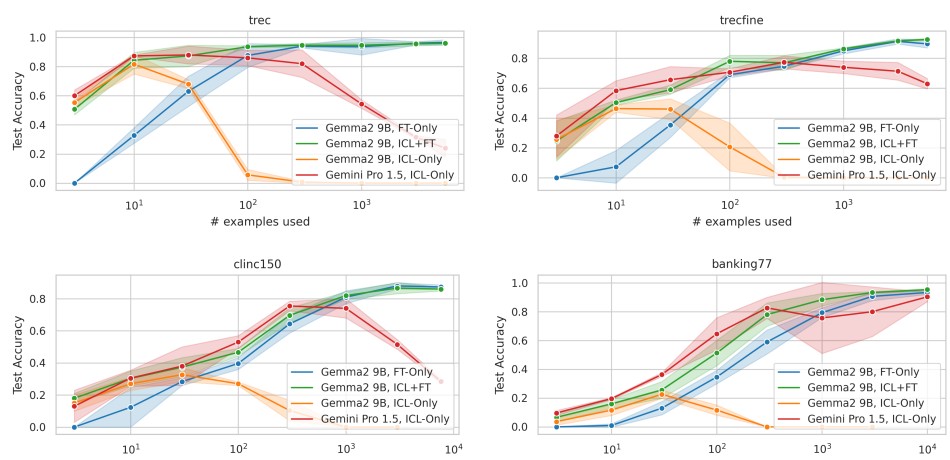

1261
1262
1263
1264
1265
1266
1267
1268
1269
1270
1271
1272
1273
1274
1275
1276
1277
1278
1279
1280
1281
1282
1283
1284
1285
1286
1287
1288
1289
1290
1291
1292
1293
1294
1295

## C.8 LoRA FINE-TUNING

We run LoRA (Hu et al., 2021) fine-tuning experiments on the full suite of BBH datasets. We chose rank 16 and apply low-rank adapters on both the feed-forward MLPs, and the KV projection matrices in the attention blocks. We use the Adam instead of Adafactor as optimizer but otherwise use the same hyperparameter-sweeps as before. Fig 17 shows the detailed results for 4 BBH tasks. The average over all BBH benchmarks is reported in Section 4.

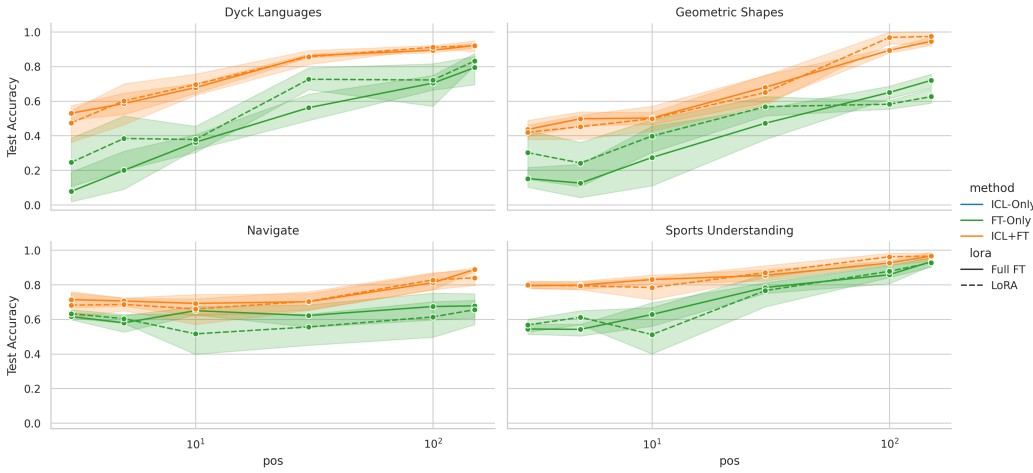

Figure 17: Test-set performance with LoRA- vs. full-finetuning for 4 representitive BBH tasks.

## C.9 PREQUENTIALLY SELECTED VS. GLOBALLY FIXED HYPERPARAMETERS

### C.9.1 GLOBALLY BEST PERFORMING HPS ON BBH

We show the average test-set accuracy over all BBH datasets and over all positions as when using globally fixed learning-rates and number-of epochs:

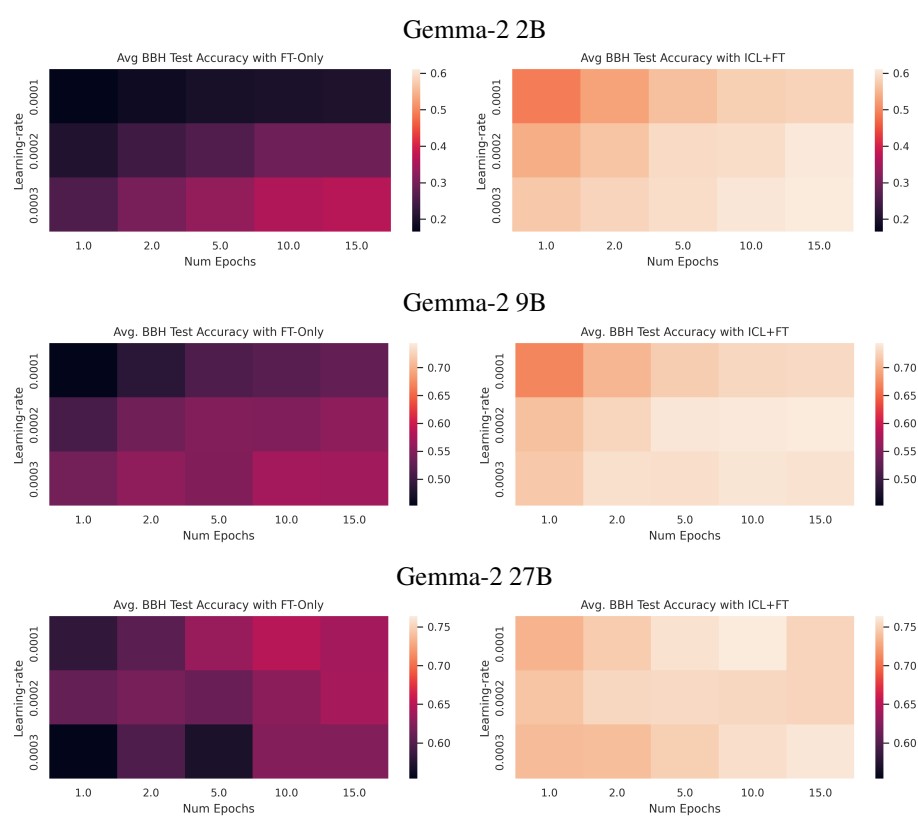

Figure 18: Avg. BBH Test Accuracy for specific hyperparameter combinations (learning-rate and number of epochs $E$).

Table 9: Best performing configurations (in respect to BBH Average Test Accuracy).

| Method | Model | learning rate | num epochs |
|--------|-------|---------------|------------|
| FT-Only | Gemma2 27B | 0.0002 | 15 |
|  | Gemma2 9B | 0.0002 | 15 |
|  | Gemma2 2B | 0.0003 | 15 |
| ICL+FT | Gemma2 27B | 0.0001 | 10 |
|  | Gemma2 9B | 0.0002 | 10 |
|  | Gemma2 2B | 0.0003 | 15 |

## C.10 COMPARISON OVER ALL BBH TASKS

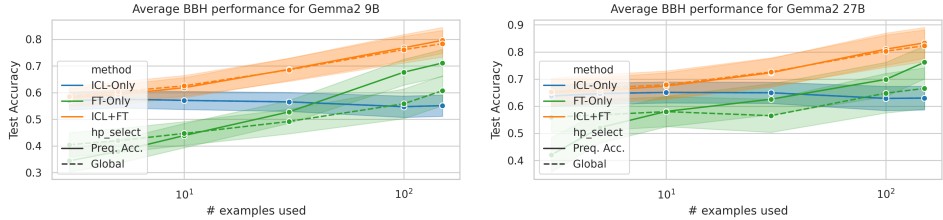

Figure 19: Average test-set accuracy over 23 BBH tasks when using either preq. hyperparameter selection or using a single fixed configuration. See 5 for comparison.

