# OpenReview forum: "Fine-Tuned In-Context Learners"
_ICLR.cc/2026/Conference — Submitted to ICLR 2026_

### Official Review · Reviewer_vqXF · 2025-10-27

**Soundness:** 4
**Presentation:** 3
**Contribution:** 2
**Rating:** 6
**Confidence:** 4

**Summary:**

The paper proposes ICL+FT, a unified adaptation method that fine-tunes LLMs with k-shot in-context (x, y) examples and then uses the in-context pairs again at inference. It selects hyperparameters via prequential next-step evaluation, eliminating the need for a held-out dev set. Across Gemma-2 model sizes and several benchmarks (e.g., BBH), ICL+FT consistently matches or outperforms ICL-only and FT-only baselines.

**Strengths:**

1. The paper is well structured and easy to follow.
2. Using prequential evaluation to bypass cross-validation significantly reduces the cost of hyperparameter selection.
3. Results span multiple datasets, including 23 BBH tasks, an NLP task suite, Parity-20, and FLoRes. This provides strong evidence that ICL+FT delivers gains over the ICL-only and FT-only baselines.

**Weaknesses:**

1. Section 3 states that prequential selection is computationally efficient, but the paper lacks explicit runtime or FLOP comparisons against CT-only, FT-only, and a simple hold-out cross-validation baseline. Concrete wall-clock measurements would help substantiate the efficiency claim.
2. Beyond ICL-only and FT-only, comparisons to other adaptation methods like prefix tuning, prompt tuning, and context tuning are missing. These would help clarify whether ICL+FT is a state-of-the-art method in performance and/or efficiency.

**Questions:**

1. How sensitive is ICL+FT to the order of training examples in Algorithm 1? Do different permutations lead to significantly different hyperparameters or final performance? How does this sensitivity compare to standard ICL-only?
2. Have you evaluated ICL+FT on models outside of the Gemma family? For example, Llama and Qwen are also open-source model families with varying model sizes.

---

> ### Author Response · Authors · 2025-11-24
>
> Thank you for you review and for the insightful comments. We believe your comments will help us improve the paper.
>
>
>
> **Computational Efficiency**
>
> We are preparing a separate common reply on computational efficiency because this question was brought up by reviewer S6Dd as well.
>
> **Additional Baselines**
>
> We considered prefix tuning as an additional parameter efficient fine-tuning method (PEFT) to investigate. However, it comes with some ambiguity about how to best combine it with ICL+FT. Ablating these different choices seemed out of scope for the current study given that we ran extensive experiments on LoRA, which is a widely used representative method for PEFT. Considering our results on LoRA, we expect that potential gains from better PEFT methods translate to gains for ICL+FT, as well.
>
> We now ran additional experiments with a simple brute-force static k-shot example selection technique: The learner maintains a small number of k-shot ICL example sets and estimates their fitness and mutates them based on incoming new ground-truth examples. With this method we see a small but statistically significant uplift on some BBH tasks for both ICL and ICL+FT. These results are consistent with the experiments from "Interaction with prompt-tuning" (lines 413ff), and Appendix C.4 (Flipped Labels) and show that both ICL-Only and ICL+FT benefit equally from prompting improvements.
>
> In summary: Our experiments suggest that methods for improved prompting, or for improved fine-tuning do not only lift the corresponding baseline performance, but also lift the performance of ICL+FT, as it naturally combines these methods.
>
> **Sensitivity to order**
>
> Thank you for this question. We investigated this carefully and found that almost all variability from the 5-differently seeded experiments comes from the order in which the ground-truth examples are presented. The confidence intervals on all plots for FT-Only, ICL-Only and ICL+FT are thus dominated by data-ordering effects.
>
> Quantifying the variance on the hyper-parameter selection is not trivial, as it depends on how many hyper-parameter combinations are under consideration. Qualitatively speaking we find that the preferred learning-rates switch rapidly and are often inconsistent across different data ordering for the first 30-100 examples before settling down and becoming more consistent for >= 100 examples on BBH. The number of preferred gradient-steps / epochs on the other hand stabilizes earlier. For ICL+FT many of the neighbouring hyperparameter configurations have quite similar test performance, while for FT-Only it varies significantly more. Overall, preferred hyper-parameters are much less stable than we are used to from large dataset (pre-)training and fine-tuning.
>
> We will add a paragraph to the paper discussing the sensitivity to data-orderings and preferred hyperparameter stability.
>
>
> **Models outside of the Gemma family?**
>
> We reimplemented our method with in pytroch and are currently running experiments with the Qwen-3 model family. We post these in a separate thread for all reviewers to see.

---

### Official Review · Reviewer_ynkh · 2025-10-31

**Soundness:** 4
**Presentation:** 3
**Contribution:** 3
**Rating:** 6
**Confidence:** 4

**Summary:**

The authors consider fine-tuning in a low-data regime. They introduce a method to fine-tune on $k$ in-context examples concatenated together with each training example and additionally design a hyperparameter selection method which is better adapted to low-data contexts. They study both their training approach and hyperparameter selection method in a variety of low-data tasks, such as Big Bench Hard and low-resource translation, among others. They compare their fine-tuning method against both fine-tuning alone as well as in-context learning alone. To study their hyperparameter selection method, they compare its performance against hyperparameter selection using an i.i.d. evaluation set as well as against using a fixed global set of hyperparameters.

**Strengths:**

The prequential hyperparameter selection algorithm is is original and novel. There is some prior work on fine-tuning with few-shot examples (some cited as well as other recent work [1]), but the contribution on this front is still novel. The paper is well-written and figures are clear.

[1] Lu, Jack, Ryan Teehan, Zhenbang Yang, and Mengye Ren. "Context Tuning for In-Context Optimization." https://arxiv.org/abs/2507.04221

**Weaknesses:**

The claim, "We emphasize that the prequential training and evaluation protocol described in Section ?? does not necessitate a separate held-out set, allowing practitioners to utilize all data points for training" is too strong and not justified by the paper. At best, the paper seems to indicate that, in the low-data regime, we do not need a separate test set for hyperparameter tuning specifically. If we want to assess overfitting and generalization, we would still need a held-out test set.

Some comparisons with baselines do not seem precisely 1-1.

Other comments:

There is a broken section reference on page 5 in the Big Bench Hard paragraph and a broken citation on page 14.

**Questions:**

Can you explain this point: "Note that globally-chosen hyper-parameters introduce information leakage as a large number of test-set examples are used to chose these"? Wouldn't the evaluation set still be held-out?

Does your FT baseline also take multiple gradient steps per example?

How do you account for the fact that the ICL+FT setting has seen some datapoints multiple times (because they later appear as ICL examples for future training steps)? Do your baselines see each example the same number of times as your method does?

---

> ### Author Response · Authors · 2025-11-24
>
> Thank you for your review, the additional reference and the concrete and insightful questions. We will try to improve the paper to address these directly for future readers.
>
>
> **Not requiring held-out sets**
>
> We stand by our claim that prequential evaluation is a statistically sound model selection method that does not require held-out sets. Its statistical properties and guarantees are slightly different from held-out validation, and are in fact related to (or rather, equivalent with) Bayesian marginal likelihood model selection. We would like to point to the literature cited in lines 140-157 for theoretical treatise of the differences.
>
> More practically speaking, and to build some intuition, we discuss overfitting: When overfitting, a training method makes a model memorize the training data unreasonably well, while at the same time harming generalization performance on unseen data. In prequential training, we alternate training on an increasing pool of training examples with evaluation on new, unseen data. Because the model is regularly tested on unseen data, and its generalization performance is assessed, overfitting will be clearly damaging the prequential next-step loss.
>
> Note that prequential losses are always computed on unseen data, and is thus different from the training loss, which would indeed not be able to identify overfitting for >1 epochs.
>
> **Globally-chosen hyper-parameters introduce information leakage**
>
> The test-set has not influenced the parameters of the different model selection candidates – in that sense it is properly held-out. However, the test-set decided which of these candidates was ultimately chosen. Effectively, we report the best performing candidate even though in a realistic setting we would not have had any data left to choose any specific candidate.
>
> As an analogy: Consider a learning algorithm where we randomly perturb some model parameters in K different ways, and then use an eval-set to select the best-performing perturbation. By repeating this procedure, we "leak" log(K) bits of information from the eval-set into the parameters with each iteration.
>
> **Does your FT baseline also take multiple gradient steps per example?**
>
> Yes, we perform between 1 and 15 gradient steps per example, essentially training with between 1 and 15 epochs.
>
> **How do you account for the fact that the ICL+FT setting has seen some datapoints multiple times?**
>
> Yes, ICL+FT sees examples more often than FT-Only and we do not actively account for it. We trust that prequential model-selection, which always measures the generalization capabilities of our models as they train, identifies the best performing hyper-parameter configuration empirically.
>
> That trust seems justified because it indeed identifies models that perform well on held-out data (see e.g. Figure 6).
>
> We did verify that increasing the number of epochs to 20 did not improve the results on any of the BBH tasks – the range of 1 to 15 epochs is thus sufficient for both FT-Only and ICL+FT;

---

### Official Review · Reviewer_S6Dd · 2025-11-03

**Soundness:** 3
**Presentation:** 3
**Contribution:** 1
**Rating:** 2
**Confidence:** 4

**Summary:**

The paper proposes a unified approach that combines in-context learning (ICL) and fine-tuning (FT) to improve task adaptation in large language models. Instead of using these methods separately, the model is fine-tuned on prompts that already include k-shot examples, and during inference it again receives k in-context examples. Technically, the ICL+FT method forms training sequences consisting of k demonstrations followed by the target query, and updates model parameters using the likelihood over all answer tokens. For hyperparameter tuning, the authors introduce a prequential evaluation scheme, which incrementally trains and evaluates the model without requiring a separate validation set. Experiments on various benchmarks show that ICL+FT typically matches or sometimes outperforms both ICL-only and FT-only baselines across Gemma 2 based models (2B, 9B, 27B).

**Strengths:**

- Clear and simple idea that is easy to implement.
- Broad empirical performance outperformed other baselines across tasks, model sizes, and data budgets.
- Useful ablations on number of in context examples, instruction prompting, and LoRA.

**Weaknesses:**

- **Conceptual novelty is limited and close to MetaICL style training.** Prior work on MetaICL and related meta learning frameworks already trains on k shot episodic inputs so that models learn to use in context examples. This paper differs mainly in scope, since it targets a single downstream task with task specific fine tuning rather than cross task generalization without parameter updates. The core learning signal of using in prompt examples is therefore very similar.
- **Limited and Unbalanced Efficiency Claims:** While the paper suggests both data and computation efficiency, this claim appears overstated. The approach still requires per-task fine-tuning and utilizes k in-context examples during inference, resulting in cumulative rather than reduced compute cost. Therefore, although data efficiency may be plausible, there is no clear evidence of computational efficiency.
- **Ambiguity in training details and sensitivity.** The algorithm samples k context examples from previously seen data, but the selection strategy and order effects aren’t analyzed. (performance of ICL is known to highly sensitive to those factors.) There is no report of variance across different context selection policies or data orderings, which can be substantial for in context methods.

**Questions:**

- Is there a reason why the title shown on OpenReview and the title in the PDF are different?

---

> ### Author Response · Authors · 2025-11-24
>
> Thank you for your review and for providing some concrete suggestions for improvement.
>
>
> **Conceptual novelty**
>
> The focus of this paper is not ICL as a general phenomenon, but rather how to adapt LLMs to specific downstream tasks when only little ground-truth training data is available. While Meta-ICL (and related work) investigates and suggests improvements for ICL in general, it is not practically applicable in these scenarios as it does not provide a recipe on how to proceed given only a few ground-truth examples.
>
> In contrast, our work here combines insights from ICL and prequential evaluation into a concrete novel algorithm. Importantly, we address the difficulty of hyperparameter selection in the small data scenario which MetaICL is unable to handle effectively. As far as we know, this algorithm attains significantly improved prediction performance compared to other approaches that can be applied in these small-data scenarios.
>
> **Efficiency**
>
> This point was brought up by a second reviewer and we are preparing a common reply. Please see the general discussion.
>
>
> **Ambiguity**
>
> Can you elaborate what you mean with "Ambiguity in training details"? We hope it is clear from Sec. 3 and Alg. 1. that we perform uniform random exemplar selection throughout the paper.
>
> **Sensitivity**
>
> There are two types of sensitivity that need to be distinguished:
> 1) Sensitivity to the exact ground-truth examples available to solve the task (i.e. the order in which these are made available).
> 2) Sensitivity to the choice of which of the available examples are used as k-shot examples for ICL-only and IFL+FT.
>
>
> Regarding 1): We carefully estimated this sensitivity by running 5 random seeds for each experiment and always visualized the 2 $\sigma$ SEM in all plots. See lines 200-206 for details.
> With SEM = $\frac{\sigma}{\sqrt 5}$ the underlying variance is $\sigma^2 = 1.25 R^2$, where R is the reported 2 $\sigma$ SEM on each datapoint. From the plots we can see that for most experiments with >=30 examples the 2 $\sigma$ SEM is below 5%, often below 1% prediction accuracy points. For the very small-data regime <=10 examples the uncertainty is sometimes >10%, usually however 5% or smaller.
>
> *Note that when comparing methods (ICL-Only vs. FT-Only vs. ICL+FT), the performance gap is usually significantly above these uncertainties; as can be seen in the plots.*
>
> We will add this uncertainty information to the text and tables to ensure this is more salient to the reader. It was indeed not obvious that our results are statistically significant.
>
>
> Regarding 2): We initially experimented with ICL example selection method from [Liu et al., 2021], however found them unreliable on BBH and providing only minor uplift even when >100 examples were available. Furthermore, these methods come with hyperparameters, and choosing them in the small data-regime proved to be finicky. We did not feel comfortable including these results without a deeper understanding of why they fail on BBH in the small-data regime, which seems outside of the scope of this study.
>
>
> We now ran additional experiments with a simple brute-force static k-shot example selection technique: The learner maintains a small number of k-shot ICL example sets and estimates their fitness and mutates them based on incoming new ground-truth examples. With this selection method we see a small but statistically significant uplift on about ⅓ of the BBH tasks for both ICL and ICL+FT. We suspect that there is more headroom for improved ICL exemplar selection in the incremental small-data learning scenario. These results are consistent with the experiments from "Interaction with prompt-tuning" (lines 413ff), and Appendix C.4 (Flipped Labels) and show that both ICL-Only and ICL+FT benefit equally from prompting improvements. We will extend these paragraphs and describe that improved techniques for exemplar selection (and better prompt-tuning in general) uplift both ICL-Only and and ICL+FT.
>
>
> [Liu et al., 2021] What Makes Good In-Context Examples for GPT-3?,

---

### Author Response · Authors · 2025-11-24
**Additional Experiments with Qwen-3 models**

We are currently undertaking additional experiments with models from the Qwen3 family to validate the generalizability of our findings beyond the Gemma architecture. For the gradient-based methods, we employ full fine-tuning with the Hugging Face transformers library, operating at bfloat16 precision and using the AdamW optimizer. For each experiment, we perform a hyperparameter sweep over learning rates ${1\text{e-5}, 2\text{e-5}, 3\text{e-5}}$ and number of epochs ${1, 3, 5, 10, 15}$, while retaining the default settings for all other hyperparameters. In the ICL+FT configuration, we consistently use $k=3$ in-context examples during both the training and testing phases.

These results are provisional, as the experiments remain ongoing. For the majority of the subsequent experiments, we have completed 3 random seeds and currently observe a 2-$\sigma$ Standard Error of the Mean (SEM) ranging from 5 to 10 percentage points across most of the reported results.


## Geometric Shapes


**Geometric Shapes, Qwen 0.6B**


|  pos | ICL-Only | FT-Only | ICL+FT |
|-----:|---------:|--------:|-------:|
|    3 |     0%   |    8%   |   13%  |
|   10 |     0%   |   15%   |   10%  |
|   30 |     0%   |   30%   |   29%  |
|  100 |     0%   |   33%   |   50%  |


**Geometric Shapes, Qwen 1.7B**


|  pos | ICL-Only | FT-Only | ICL+FT |
|-----:|---------:|--------:|-------:|
|    3 |    19%   |   18%   |   27%  |
|   10 |    20%   |   26%   |   45%  |
|   30 |    15%   |   31%   |   54%  |
|  100 |    40%   |   52%   |   79%  |


**Geometric Shapes, Qwen 4B**


|  pos | ICL-Only | FT-Only | ICL+FT |
|-----:|---------:|--------:|-------:|
|    3 |    39%   |  25%   |   50%  |
|   10 |    42%   |  23%   |   47%  |
|   30 |    53%   |  40%   |   70%  |
|  100 |    61%   |  52%   |   78%  |



With Qwen3 30B-A3B ICL-Only we obtain 40%, 48%, 46% and 65% for 3, 10, 30 and 100 examples respectively (all +/- 10% 2-sigma uncertainty).




## Sports Understanding


**Sports Understanding, Qwen3 0.6B**


|  pos | ICL-Only | FT-Only | ICL+FT |
|-----:|---------:|--------:|-------:|
|    3 |    0.0%  |    0%   |   49%  |
|   10 |    0.0%  |   15%   |   42%  |
|   30 |    0.0%  |   18%   |   46%  |
|  100 |    0.0%  |   48%   |   43%  |


**Sports Understanding, Qwen3 1.7B**


|  pos | ICL-Only | FT-Only | ICL+FT |
|-----:|---------:|--------:|-------:|
|    3 |    31%  |    7%   |   50%  |
|   10 |    29%  |   20%   |   57%  |
|   30 |    38%  |   46%   |   61%  |
|  100 |    43%  |   51%   |   58%  |


**Sports Understanding, Qwen3 4B**


|  pos | ICL-Only | FT-Only | ICL+FT |
|-----:|---------:|--------:|-------:|
|    3 |     36%  |    8%   |   64%  |
|   10 |     38%  |   28%   |   68%  |
|   30 |     36%  |   55%   |   63%  |
|  100 |     40%  |   54%   |   68%  |


With Qwen3 30B-A3B ICL-Only we obtain 38%, 39%, 39% and 40% for 3, 10, 30 and 100 examples respectively (all +/- 10% 2 sigma uncertainty).

---

### Meta-Review · Area_Chair_q4b6 · 2025-12-28

**Summary:**

The paper proposes ICL+FT, a unified adaptation method that fine-tunes LLMs with k-shot in-context (x, y) examples and then uses the in-context pairs again at inference. It selects hyperparameters via prequential next-step evaluation, eliminating the need for a held-out dev set. Across Gemma-2 model sizes and several benchmarks (e.g., BBH), ICL+FT consistently matches or outperforms ICL-only and FT-only baselines.

**Reviewer Concerns:**

Advantages:
1. The paper is well structured and easy to follow, with a clear and simple idea that is easy to implement.
2. Broad empirical performance outperformed other baselines across tasks, model sizes, and data budgets. Using prequential evaluation to bypass cross-validation significantly reduces the cost of hyperparameter selection.
3. Useful ablations on number of in context examples, instruction prompting, and LoRA.

Disadvantages:
1. Limited novelty. The reviewer argued that this work is close meta-ICL style training.
2. Some claims are too strong or not supported by what presented, particularly regarding the computation efficiency (cumulative rather than reduced compute cost). Reviewer also challenged on that prequential selection is computationally efficient.

3. The work missing the base models from Qwen or GLM or others (only Gemma used) and comparison to other adaptation methods (eg. prefix tuning, prompt tuning, and context tuning). Although the authors provided ongoing results, but there are significant revision required, so another round of polish is definitely necessary.

**Reviewer Scores:**

the review rating were 2,6,6,  Although the authors provided ongoing results, but there are significant revision required, so another round of polish is definitely necessary.

---

### Decision · Program_Chairs · 2026-01-26

Reject